# Feasibility, efficacy, and perceptions of an online writing intervention in patients with depressive disorders: A randomized, multi-methods pilot study

Marta M. Maslej[1,2], Abigail Ortiz[2,3], Paul W. Andrews[4], Benoit H. Mulsant[2,3,5]*

**1** Krembil Centre for Neuroinformatics, Centre for Addiction and Mental Health, Toronto, Ontario, Canada, **2** Department of Psychiatry, Temerty Faculty of Medicine, University of Toronto, Toronto, Ontario, Canada, **3** Campbell Family Mental Health Research Institute, Centre for Addiction and Mental Health, Toronto, Ontario, Canada, **4** Department of Psychology, Neuroscience & Behaviour, McMaster University, Hamilton, Ontario, Canada, **5** The Royal Ottawa, Ottawa, Ontario, Canada

* benoit.mulsant@utoronto.ca

**Data availability statement:** The minimal data for this study cannot be made openly-available, because we do not have ethical approval to share personal identifiable, patient-level data outside of the research study. Data are available

## Abstract

Barriers to accessing treatment for Depressive Disorders highlight a need for scalable interventions. Expressive writing (EW), which involves daily writing about a personal issue, has been shown to alleviate depressive symptoms in various samples. Its simple format makes it amenable to online administration. We conducted a multi-methods study to evaluate the feasibility, efficacy, and perceived utility of online EW in patients with Depressive Disorders. We recruited 53 patients diagnosed with a Depressive Disorder from a psychiatry hospital. Half were randomized to complete four 20-minute sessions of EW online and the other half were randomized as controls. All participants completed measures before and after the intervention, and at a one-month follow up. Our primary outcome for feasibility was the number of participants completing at least three sessions of EW, and for efficacy, the primary outcome was depressive symptom severity. Secondary outcomes were anxiety symptom severity, functional impairment, and perceived personal problem complexity. Mixed effect models assessed the impact of condition (EW or control), time, and their interaction on each outcome. Qualitative interviews with 11 participants assigned to EW probed into their experiences, which were subject to reflexive thematic analysis. Most EW participants (73%) completed at least three EW sessions, but only 23% wrote for 20 minutes. There was no evidence that condition interacted with time to impact depressive symptoms ($F = 0.10$, $p = .904$), anxiety symptoms ($F = 0.17$, $p = .847$), functional impairment ($F = 1.03$, $p = .363$), or perceived problem complexity ($F = 0.94$, $p = .394$). In qualitative interviews, some participants reported EW allowed them to offload or process negative emotions and thoughts, whereas others reported it to be unhelpful. EW was feasible to administer online to patients with Depressive Disorders, with some reported benefits for emotional and cognitive processing. However,

from the study's first author, Marta Maslej, Krembil Centre for Neuroinformatics, Centre for Addiction and Mental Health, marta.maslej@camh.ca, for researchers who meet the criteria and secure the necessary approvals for accessing the confidential data. Code used to generate results is openly-available on the OSF (https://osf.io/wqpge).

**Funding:** This work was supported by a Discovery Fund (DF) from The Centre for Addiction and Mental Health (to MMM) and a Health Systems Impact Fellowship (HSIF) from Canadian Institutes of Health Research (CIHR) (to MMM) and the Labatt Family Chair in Biology of Depression in Late-Life Adults at the University of Toronto (to BHM). The funders had no role in study design, data collection and analysis, decision to publish, or preparation of the manuscript.

**Competing interests:** The authors have declared that no competing interests exist.

EW was not associated with improvements in symptoms and may not be an effective online intervention for Depressive Disorders in its current state. Providing more guidance, with some clinical oversight or support, may be a promising approach to modifying the intervention in future work. **Trial registration**: ClinicalTrials.gov NCT06699719

## Introduction

Depressive Disorders, such as Major Depressive Disorder (MDD) and Persistent Depressive Disorder (PDD), are common and distressing mental health conditions, which are often left untreated. An estimated 11–17% of Canadians will be diagnosed with MDD in their lifetime [1,2], whereas the number of affected individuals has increased due to the COVID-19 pandemic. Since its onset, the percentage of Canadians reporting high levels of depression has increased from 4 to 10%, with even larger increases reported for anxiety (from 5 to 20%) [1]. Psychotherapy is an effective and enduring treatment option [3], but its availability is limited. Despite an increase in psychotherapy provision over the past decade, many Canadians still cite financial and access issues as barriers [4–6]. Recent shifts to virtual care [7] have not addressed this unmet need. Canadians feeling depressed or anxious report that the quantity and quality of mental health services has declined since the onset of the pandemic [1], potentially due to staff shortages and health system burden. Surveys indicate that over one quarter of Canadian adults are now unable to manage their sadness, anxiety, or stress, and 53% of those seeking mental health care do not receive it [7]. This leaves an estimated 5.2 million Canadians grappling with distressing symptoms and related difficulties in social, occupational, and physical functioning [7,8]. There is a clear need for interventions for Depressive Disorders that can alleviate symptoms and improve functioning, while being accessible and scalable, i.e., not requiring intensive psychotherapist involvement.

One possible intervention is Expressive Writing (EW), which involves a series of daily journaling tasks in which individuals write their deepest thoughts and feelings about a negative event or problem for at least 20 minutes [9]. EW does not require manuals or trained clinicians, making it affordable and amenable to being administered online. It has been associated with various physical and psychological benefits, which have been linked to several proposed therapeutic mechanisms. Repeated exposure to negative thoughts and feelings during EW may diminish their impact over time, [10] or lead to a realization that they are manageable and even malleable [11]. Temporary increases in emotional distress during EW may also underlie emotional and cognitive processing, which is another proposed mechanism. Translating abstract emotions and thoughts into language requires individuals to select, organize, connect, and reframe their experiences, which can lead to new perspectives or an enhanced understanding of the self [11]. These insights or resulting changes in behaviour may improve an individual's circumstances and reduce distress over time [11–14].

The evidence on the extent to which EW impacts depressive symptoms and psychological wellbeing is mixed, which may be related to the use of predominantly healthy, non-clinical samples in EW research. Some meta-analyses find small but significant net effects for alleviating symptoms of depression and anxiety [10,11,15], whereas others find no benefits for psychological wellbeing [10,16,17]. One meta-analysis of 39 randomized controlled trials of the effect of EW on depressive symptoms by Reinhold and colleagues (2018) found a small but significant decrease in symptoms immediately following EW; however, decreases in symptoms at subsequent follow-ups were not statistically significant [18]. Although this finding led the authors to conclude that EW may not be helpful in the long term, large sample variances suggested that some individuals benefit whereas others do not [18]. Another meta-analysis of 31 randomized controlled trials by Guo and colleagues (2023) examined effects of EW on depression, anxiety, and stress; their results also revealed small and significant decreases in symptoms conversely emerging at 1–3 month-follow-ups, but not immediately after writing [11]. However, most studies of EW on depressive symptoms have involved homogeneous samples largely consisting of college students and few have examined effects in patients formally diagnosed with a Depressive Disorder (e.g., comprising only 6% of studies in Reinhold and colleagues' meta-analysis) [11,18]. Two studies examining the impacts of EW on depressive symptoms involved participants who may have had Depressive Disorders, but they did not receive a formal clinical diagnosis: one study involved outpatients undergoing psychotherapy who "self-reported their primary presenting problem", which included depression [19], and the other study recruited visitors to an informational website about mood disorders (i.e., depression and bipolar disorder) [20]. Other studies have involved at-risk samples, such as college students with a history of self-reported depression or trauma [21,22]. Only one published study examined the effect of EW in a sample formally diagnosed with a Depressive Disorder, administering the intervention in-person for 20 minutes over three consecutive days in participants diagnosed with MDD experiencing a current major depressive episode based on a structured interview [23]. Compared to participants who completed control writing tasks, participants assigned to EW had fewer depressive symptoms following the intervention, and this benefit persisted after one month [23]. These changes were also associated with large effects, and they were clinically meaningful [23]. Although the degree of symptom severity required for formal diagnosis of MDD is not well defined, findings from several other studies also suggest that individuals experiencing severe and debilitating symptoms may benefit from EW more than individuals whose symptoms are minimal [20,21,24,25]. These findings highlight a need for examining the effects of EW in samples with severe symptoms of depression, or those formally diagnosed with a Depressive Disorder.

At the same time, there are concerns that administering EW to patients formally diagnosed with Depressive Disorders may be unhelpful or impractical. Despite some studies suggesting that individuals with severe symptoms can benefit from EW, this is not always the case. EW has been associated with increases in depressive symptoms or emotional distress over the long term in some samples, including those more likely to ruminate or experience severe symptoms at baseline [26,27]. These findings have led some authors to conclude that EW may be unhelpful for Depressive Disorders, because the intervention offers little guidance for processing challenging material and may exacerbate negative emotions or thoughts [26,27]. Another concern is related to feasibility. Low motivation or other depressive symptoms may prevent individuals from engaging with EW or completing it as instructed, particularly if it is administered remotely. Although no studies to-date have administered EW online to patients diagnosed with Depressive Disorders, rates of attrition in participants assigned to complete EW remotely are generally high [27,28], with life commitments and difficulties engaging with the task cited as barriers [20]. Thus, it is unclear whether patients diagnosed with Depressive Disorders (e.g., MDD, PDD) would engage with EW administered remotely and if so, whether they would benefit from it [23].

Benefits of EW may not only reflect changes in symptoms, but they may extend to other outcomes. The proposed therapeutic mechanism of EW involves emotional and cognitive processing [9–11], suggesting that individuals with Depressive Disorders may find EW valuable for thinking through difficult experiences or gaining a different understanding of events, thoughts, and feelings. These outcomes of EW may be better captured with qualitative methods. In one study, for example, EW about the COVID-19 pandemic did not impact depressive symptoms as assessed via a structured questionnaire,

but two thirds of participants interviewed about EW reported it to be personally beneficial [29]. Emotional or cognitive processing during EW may also be associated with changes in personal, social, or occupational functioning, which may explain noteworthy benefits observed in some EW studies. For example, unemployed participants were more likely to find a job [30], or participants struggling with marital separation either re-united with an ex-partner or reported increased emotional detachment from them [31]. Several studies have also observed higher grade point averages for students who wrote expressively about an upcoming exam or adjustment to college [32,33]. Given that depressive symptoms do not always map onto personal or functional outcomes [8], it is possible that in studies not reporting statistically significant effects of EW on depressive symptoms [18], participants might nevertheless experience benefits related to other aspects of their lives. EW may help participants reconceptualize or even address their personal problems or improve their physical, social, or occupational functioning, making it important to measure other outcomes of EW, like personal functioning or perceived problem complexity.

## Aims and hypotheses

We conducted a randomized, multi-methods study to assess the feasibility and efficacy of administering EW online to patients with Depressive Disorders (MDD, PDD, or both), as well as their perceptions of its impact on emotional and cognitive wellbeing. To achieve these aims, we randomized patients into one of two groups: one group was prompted to complete four sessions of EW online, whereas the other group was not asked to complete this task. As employed in some prior studies of EW [11,23], we did not administer a control writing task (i.e., writing about non-emotional topics) due to our focus on feasibility. Our aim was to examine whether a group of patients with MDD would complete a writing intervention online, which requires a comparison of their study completion rates and outcomes to a group not asked to complete any writing or intervention at all. We examined the rate of EW completion and adherence to the suggested writing duration. Based on the prior study of EW in participants with MDD [23], we defined feasibility as 70% of EW participants completing at least three 20-minute sessions of writing.

To examine the impacts of EW on outcomes, both groups completed self-reported measures. Our primary outcome was reduction in depressive symptoms; secondary outcomes included symptoms of Generalized Anxiety Disorder (GAD), functional impairment, and the perceived complexity of personal problems. Participants completed these measures before and after the EW task, and at a one-month follow up. We expected different changes in outcomes across time for both groups. In the control group, we anticipated no change or a slight decrease in symptoms of MDD/PDD and GAD, functional impairment, and perceived problem complexity over time. In the EW group however, we considered three possible outcomes immediately following the intervention relative to the control group: a significant reduction in MDD/PDD and GAD symptoms [18,23], no differences in these symptoms [11], or increased symptoms, reflecting temporary increases in emotional distress [12,13]. Given that changes in functional impairment or perceived problem complexity may take some time to emerge, we did not expect effects related to these outcomes immediately following EW. At the one-month follow up, however, we expected the EW group to report fewer symptoms of MDD/PDD and GAD, less functional impairment, and less complex personal problems, relative to the control group. We also conducted qualitative interviews to probe into participants' experiences with EW. We expected participants would report EW to be beneficial, particularly as related to its emotional or cognitive effects [10,29].

## Materials and methods

### Recruitment

We recruited outpatients from the Centre for Addiction and Mental Health, a large academic psychiatry hospital located in Toronto, Ontario, from January 23, 2018 to June 21, 2022. Patients were referred to the study by psychiatrists during initial or follow-up appointments, or by psychotherapists either before the start or after the end of group Cognitive Behavioral

Therapy. The psychiatrists or psychotherapists were asked to refer patients who have been diagnosed with MDD. Inclusion criteria were adults aged 18 or older, who were assessed by a psychiatrist and received a diagnosis of MDD or another Depressive Disorder (e.g., PDD) with or without another co-morbid condition. Participants also required access to the internet and an email address at which they could receive study materials. Patients were excluded if they could not understand, speak, or write in English, which were criteria determined based on self-report at study enrollment. We did not have access to translation services, so these criteria ensured participants could provide informed consent and complete questionnaires and qualitative interviews. It also enabled us to screen EW texts for evidence of suicide risk (see S1 File for the study's suicide risk management protocol). Clinicians were also asked not to refer patients into the study if they had severe cognitive impairments or very limited literacy, which would prevent them from being able to complete a writing task. Although we had a protocol in place for following up with participants who reported suicidal ideation in the study (S1 File), we asked clinicians not to refer patients reporting suicidal thoughts and behaviours or other mental health symptoms which the clinicians felt could put them at risk for negative outcomes when completing the intervention remotely. However, we did not formally assess patients for these latter two criteria upon enrollment into the study. In this pilot feasibility study, we also did not formally assess patients for a diagnosis of Depressive Disorder with a structured, clinical interview. Patients were not excluded if they were receiving treatment for MDD (e.g., antidepressants, group psychotherapy).

**Measures**

**Demographics.** Participants self-reported their demographic characteristics, which included their age, gender, racial or ethnic background, and postal code, which was used to derive a measure of neighbourhood income level as a proxy of participant socioeconomic status. We matched clients' postal codes with average incomes for their dissemination areas, using national Census statistics. Incomes represented across all provincial dissemination areas were split into four categories (i.e., low, mid-low, mid-high, high), to which incomes for individual participants were matched.

**Patient Health Questionnaire (PHQ-9).** The PHQ-9 is a self-reported measure of depressive (i.e., MDD/PDD) symptom severity validated for use in psychiatric samples [34,35] and in Canadian populations [36] It measures the frequency of 9 symptoms on a scale from 0 (*not at all*) to 3 (*nearly everyday*). Scores are summed to reflect depressive symptom severity, and they range from 0-27, with higher scores indicating greater severity.

**Generalized Anxiety Disorder (GAD-7).** The GAD-7 is a self-reported measure of GAD symptom severity [37]. It has good psychometric properties [38] in measuring the frequency of 7 symptoms on a scale from 0 (*not at all*) to 3 (*nearly everyday*), and it has been used to assess GAD in Canadian samples [39] Summed scores range from 0-21, with higher scores indicating greater severity.

**WHO-Disability Assessment Schedule (WHODAS).** WHODAS is a self-reported measure of functional impairment [40]. It measures the degree of difficulty from the past month related to 12 activities in domains of cognition, mobility, self-care, and social participation. The WHODAS has been validated for measuring mental health-related disability in Canadian community samples [41]. Activities are measured on a scale from 0 (*none*) to 4 (*extreme or cannot do*), and scores are summed to reflect overall functional impairment. They range from 0-48, with higher scores indicating greater impairment.

**Problem Complexity Questionnaire (PCQ).** PCQ measures the perceived complexity of personal problems. It consists of 8 statements reflecting the complexity of problems, such as "I do not yet know how to resolve these problems" or "These problems have left me in a dilemma". Participants rate each item on a scale from 1 (*completely disagree*) to 4 (*completely agree*). Scores range from 8 to 32, with higher scores indicating higher perceptions of problem complexity. The PCQ has been used to measure perceived problem complexity in Canadian samples [42], and it has undergone psychometric testing in a sample with MDD, demonstrating an acceptable mean inter-item correlation and excellent internal consistency [43].

## Ethics statement and data availability

All procedures relating to this research were approved by the hospital's Research Ethics Board (REB# 100/2018), and they were carried out in accordance with its ethical standards and with the Helsinki Declaration. All participants gave written informed consent. The study is registered with ClinicalTrials.gov (NCT06699719). Due to our focus on the feasibility of administering the intervention online (i.e., to determine if participants would complete at least 3 sessions of writing), we did not register our study as a randomized trial prior to the enrollment of the first participant. The CONSORT table and trial protocol are provided in S1 File.

## Procedures

Patients referred to the study were provided with a high-level description of study aims (i.e., "evaluating the extent to which completing questionnaires or a writing task is useful for people with depression"). Prior to enrolment, all participants were given information about study procedures and risks, and they provided their informed consent. Once enrolled, participants were allocated to EW or control (1:1 allocation ratio) with 4-block randomization and they were sent the first study questionnaire consisting of the demographics form, PHQ-9, GAD-7, WHODAS, and PCQ. Their electronic health records were also accessed to extract information about their diagnoses and current treatment. After completing the first questionnaire, participants assigned to EW were sent instructions to complete the first EW session on the next day. We adapted a common version of EW instructions [44], also used in prior research on EW in samples with MDD [23], which reads:

*For the next 20 minutes, please use the box below to write your very deepest thoughts and feelings about a negative issue or personal problem that you are currently being affected by. In your writing, try to let go and explore your emotions and thoughts regarding this issue or problem. Please do not worry about spelling or grammar. You may wish to set a timer or have a clock handy to make sure you write for 20 minutes.*

Instructions were accompanied by a text box, in which participants could write freely. Although participants were instructed to write for 20 minutes, they were also told during the informed consent discussion that they could pause or stop the writing task at any time (e.g., if they felt unwell). EW participants were provided with the same instructions on the next three days. Participants who were allocated to control were not provided with the EW instructions. After one week (for the control group) or on the next day that participants completed their fourth EW session (for the EW group), participants received a second questionnaire containing the PHQ-9, GAD-7, WHODAS, and PCQ. One month after completing the second questionnaire, they received a third questionnaire containing the same measures. All study materials (i.e., questionnaires, EW sessions) were administered via RedCap, a web-based application for data collection available through the hospital, and instructions were sent to participants by email. If participants did not complete an EW session or follow-up questionnaire within three days of receiving it (or after one week for the first questionnaire), participants who agreed to receive reminders by text message or email were sent up to two reminders. Participants who did not respond or complete the relevant study components after two reminders were assumed to have dropped out of the study and were no longer contacted. However, participants who responded to reminders or completed the questionnaires or EW sessions within any period of time were included.

After completing the third questionnaire, all EW participants were invited to participate in a brief qualitative interview. Qualitative interviewing was guided by the question of how individuals with Depressive Disorders experience online EW, with a focus on understanding its emotional and cognitive effects (see S2 File for the interview guide). At this point, control participants were given an opportunity to complete the EW task, if they wished. All participants who completed the study were sent a small gift card by mail. The study's first author was responsible for implementing randomizing, enrolling participants, assigning participants to interventions, and administering study questionnaires.

## Statistical analysis

**Descriptive.** We examined the baseline demographic and clinical characteristics of our sample and compared them between the control and EW groups. Differences in categorical variables were compared with Chi-squared tests. Due to

small samples in some racial or ethnic categories, participants were grouped into one of two categories when testing for differences between the control and EW groups: white or racially marginalized; this latter category included participants identifying as First Nations, West/East/South Asian, Latin American, Mixed Race, or Other (see Table 1). Continuous variables that were normally distributed were compared with t-tests, whereas Wilcoxon-signed rank tests were used for continuous variables with non-parametric distributions.

**Feasibility and efficacy.** We examined how many EW participants completed at least three sessions, how many wrote for at least 20 minutes, how many required reminders, the median length of time spent writing, the median word count of EW tasks, and average time intervals between writing sessions.

In our intention-to-treat analysis, we generated mixed effect models with maximum likelihood estimation using all participants in the EW group and all participants in the control group who completed the first (baseline) questionnaire. We generated four models to examine the impact of condition, time, and a condition by time interaction on each outcome (i.e., PHQ-9, GAD-7, WHODAS, PCQ). Time was treated as a categorical factor, to account for the possibility of non-linear effects. For any models with evidence of this interaction, we plotted estimated means to inspect trajectories, with 95% confidence intervals to gauge effects. In a per-protocol analysis, we generated the same mixed effect models adding number of completed sessions as a predictor, to determine whether completing fewer sessions impacted outcomes (setting all

**Table 1. Baseline clinical and sociodemographic characteristics of all participants, and stratified by group.**

|  | All | EW | Control | Comparison | Qualitative |
|---|---|---|---|---|---|
| *N* | 53 | 26[a] | 27 |  | 11 |
| Age, M (SD) | 36.86 (14.00) | 37.91 (11.97) | 35.93 (15.76) | $W = 283.5, p = .450$ | 41 (14.11) |
| Gender, *n* (%) |  |  |  |  |  |
| Men | 17 (32) | 6 (23) | 11 (41) | $X^2 (1) = 0.98, p = .322$ | 3 (27) |
| Women | 35 (66) | 19 (73) | 16 (59) |  | 8 (73) |
| Ethnicity, *n* (%) |  |  |  |  |  |
| White | 35 (66) | 16 (62) | 19 (70) | $X^2 (1) = 0.04, p = .847$[b] | 9 (82) |
| Racially marginalized | 17 (32) | 9 (35) | 8 (30) |  | 2 (18) |
| First Nations | 1 (2) | 0 (0) | 1 (4) |  | 0 (0) |
| West Asian | 3 (6) | 2 (8) | 1 (4) |  | 0 (0) |
| East Asian | 1 (2) | 1 (4) | 0 (0) |  | 1 (9) |
| Latin American | 3 (6) | 2 (8) | 1 (4) |  | 0 (0) |
| South Asian | 5 (9) | 2 (8) | 3 (11) |  | 0 (0) |
| Mixed | 3 (9) | 2 (8) | 1 (4) |  | 1 (9) |
| Other | 1 (2) | 0 (0) | 1 (4) |  | 0 (0) |
| Income, *n* (%) |  |  |  |  |  |
| High | 21 (40) | 8 (31) | 13 (48) | $X^2 (3) = 11.32, p = .010$ | 4 (36) |
| Mid-high | 12 (23) | 11 (42) | 1 (4) |  | 5 (45) |
| Mid-low | 6 (11) | 2 (8) | 4 (15) |  | 1 (9) |
| Low | 14 (26) | 5 (19) | 9 (33) |  | 1 (9) |
| PHQ-9, M (SD) | 14.63 (6.32) | 13.29 (6.07) | 15.82 (6.53) | $t (49) = 1.43, p = .159$ | 14.90 (4.53) |
| GAD-7 | 10.92 (6.27) | 10.25 (6.56) | 11.52 (6.05) | $W = 357.5, p = .533$ | 10.73 (5.93) |
| WHODAS | 17.96 (9.34) | 16.75 (9.28) | 19.04 (9.43) | $t (48) = 0.87, p = .387$ | 18.36 (8.20) |
| PCQ | 29.96 (5.29) | 29.46 (6.50) | 30.48 (3.72) | $W = 271, p = .923$ | 30.55 (5.28) |

[a]One participant in the EW condition did not report their age, gender, and race/ethnicity.

[b]The comparison between conditions for race/ethnicity was based on a binary representation (i.e., white vs. racially marginalized).

control participants to 0). As an additional per-protocol analysis, we replaced session number with text length (i.e., word counts) over the total number of sessions as a predictor, to determine whether writing more content impacted outcomes. Alpha level for determining statistical significance was 0.05, and residuals for each model were inspected to ensure no violations in the normality assumption. Analyses were completed in R (Version 4.1.1), using the *lmer* package [45] for mixed effect modelling.

**Experience.** Our approach to analysing qualitative interviews was guided by reflexive thematic analysis [46,47], which is a recursive and constructive process of identifying and analysing themes. Our data item was the content from qualitative interviews with participants assigned to EW and agreeing to be interviewed. The process of qualitative analysis involved multiple stages. First, we gained familiarity with the data item by reading each interview and making initial notes or ideas for coding. Next, we generated initial codes of content features, by re-reading the data item with the aim of identifying as many codes (features, themes, or patterns) as possible. We examined relationships among codes to generate broader themes, which we then reviewed and revised based on additional re-readings of the interview and recoding of missed features. We continued to refine the themes until additional changes were not leading to improvements of the thematic structure. Finally, we defined and named the resulting themes. Given our focus on emotional and cognitive effects, we adopted a predominantly theoretical or deductive approach to this process of analysis, to provide a nuanced account of how participants experienced these effects of EW. To accommodate the diversity of experiences, we incorporated inductive (data-driven) analysis [47]. Our analysis was primarily semantic and essentialist, since we wished to understand participants' explicit experiences or accounts of EW, rather than deriving latent themes. Coding and analysis of qualitative interviews was carried out by a single study team member, and it was supported by Nvivo (Version 1.7).

## Results

### Descriptive

A sample of 63 patients consented to participation, were enrolled into the study, and randomized to EW or control groups. Fifty-four participants completed the baseline questionnaire, and 46 completed the study. We could not confirm a diagnosis of Depressive Disorder (MDD or PDD) in one patient who was randomized to the EW intervention (as described below). This patient was excluded from the analysis, leaving a final sample of 53 participants who completed the baseline questionnaire, and 45 who completed the study. A Consolidated Standards Of Reporting Trials (CONSORT) diagram of the recruitment and enrollment process is provided in Fig 1.

The health records of one participant referred to the study and randomized to EW did not include a diagnosis of Depressive Disorder; although this participant completed the study, they were excluded from the analyses and therefore are not represented in the CONSORT diagram.

Diagnosis and treatment information extracted from health records for 53 participants who completed the baseline questionnaire are provided in S3 File (one participant from the initial sample of 54 participants did not consent to our review of their health records). Of the 53 remaining participants, 41 (77%) participants had a diagnosis of MDD, 10 (19%) had a diagnosis of PDD, and one was given a diagnosis (by a psychiatrist) of "depression" with a PHQ-9 score of 24; this latter participant was retained in our analysis with a diagnosis of Depressive Disorder Not Otherwise Specified (NOS). One participant with a diagnosis of GAD did not have a Depressive Disorder, and they were excluded from the analysis. The participant who did not consent to our review of their health records was retained in the analyses with a diagnosis of Depressive Disorder NOS. Most participants had multiple recorded diagnoses (see S3 File). Most participants (53%) were also on psychotropic medications, 28% had completed psychotherapy prior to starting the study, and 11% were in psychotherapy during the study. Participants had a mean age of 37 (*SD* = 14). Most (66%) were women, and 66% of participants self-identified as white. Most participants (40%) lived in areas with incomes in the highest category. Detailed descriptive statistics and relevant between-group comparisons for the demographic and clinical characteristics of the 53 participants

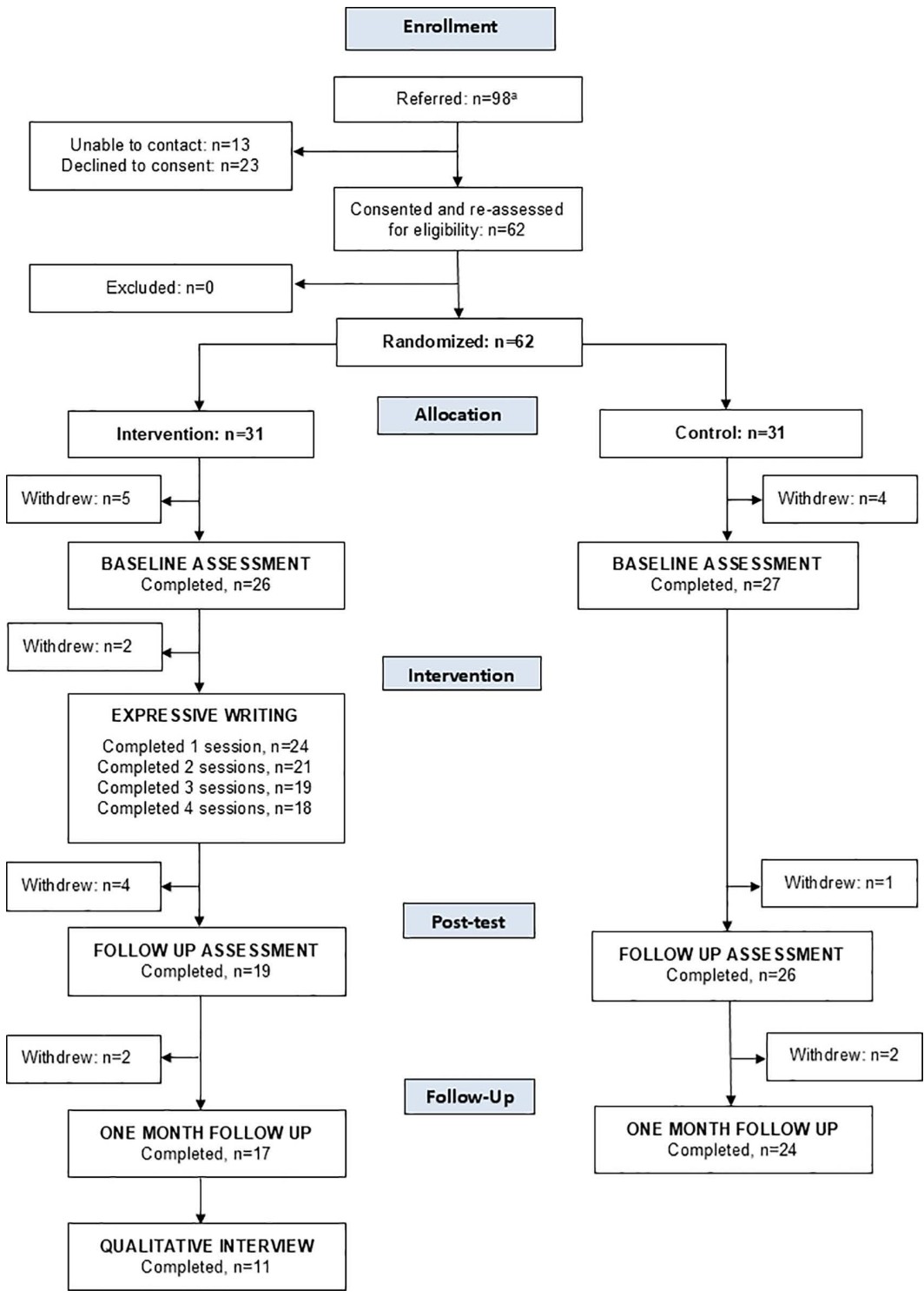

**Fig 1. A Consolidated Standards Of Reporting Trials (CONSORT) diagram of the recruitment and enrollment process.**

included in the analyses are provided in Table 1. Comparisons show that participants assigned to the EW group had higher incomes.

Table 2 provides information on the rate of EW completion (including reminders), days elapsed between sessions, writing duration, and length of the EW texts (i.e., word counts). Of the 26 participants assigned to EW, 19 (73%) completed at least 3 sessions. Most participants (88%) required at least one reminder to complete their first writing session, although fewer participants (<30%) required reminders at subsequent writing sessions. After receiving reminders, participants completed writing sessions within 2.3 days on average (ranging from 0 to 16 days). Two (7%) participants dropped out before completing the first session, one of whom indicated that they preferred not to complete EW, believing it might be detrimental to their mental health. Three (12%) participants completed only one session of writing, two (8%) completed two sessions, one (4%) completed three sessions, and 18 (69%) completed all four sessions. However, many participants did not write for at least 20 minutes, and the number of participants who wrote for at least 20 minutes decreased over subsequent sessions (see Table 2). Only 6 (23%) participants completed at least 3 sessions of EW for at least 20 minutes. The median writing duration across all sessions was 21 minutes, with the shortest writing session lasting 5 minutes and the longest lasting 4.32 days (since participants were free to leave their EW sessions open and return to them after a few days). The length of EW texts (i.e., word counts) ranged from 75 to 1,219 words. Many participants also did not write on consecutive days. Of the 21 participants who completed at least two EW sessions, mean time intervals between sessions across participants ranged from 1–21 days, with a median of 2.73 days. Only 10 (48%) participants wrote on consecutive days or had mean intervals of fewer than 2 days between sessions, whereas 6 (29%) participants had a mean interval of at least one week between sessions, or more than 6 days.

### Efficacy

**Intention-to-treat analyses.** There was no main effect of time ($F=2.09$, $p=0.131$) or condition ($F=2.48$, $p=0.121$), and no evidence of their interaction ($F=0.101$, $p=0.904$) on PHQ-9 scores. The adjusted intraclass correlation coefficient (ICC), which indicates the association of scores within each participant was 0.79. Similarly, there was no effect of time ($F=2.62$, $p=0.078$) or condition ($F=1.365$, $p=0.248$), and no evidence of their interaction ($F=0.17$, $p=0.847$) on GAD-7 scores (ICC$=0.83$). There was also no effect of time ($F=1.35$, $p=0.264$) or condition ($F=2.16$, $p=0.148$), and no evidence of their interaction ($F=1.03$, $p=0.363$) on WHODAS scores (ICC$=0.87$). Finally, PCQ scores were not impacted by time ($F=0.77$, $p=0.467$), condition ($F=0.01$, $p=0.911$), or their interaction ($F=0.94$, $p=0.394$; ICC$=0.44$). According to a *post-hoc* power analysis (reported in S4 File), our sample size of 23 participants per condition was powered to detect large effects ($f=0.47$).

**Per-protocol analyses.** The number of sessions ranged from 0-4, with 29 participants not completing any sessions, including 27 control participants and the two EW participants who withdrew prior to completing the first writing session. Word counts across all EW sessions for each participant ranged from 0 (for control participants) to 3,440 words

**Table 2. Descriptive information on EW tasks.**

| Session | Reminded[b] n (%) | Completed n (%) | Since last session (days) Median (Range) | 20 minutes n (%) | EW writing duration (minutes) Median (Range) | EW text length (words) Median (Range) |
|---|---|---|---|---|---|---|
| 1 | 23/26 (88%) | 24 (92%) | – | 18 (69%) | 22 (10–2166 or 1.5 days) | 560 (171-1219) |
| 2 | 1/24 (4%) | 21 (81%) | 1.16 (0.05-21.32) | 11 (42%) | 22 (7–6227 or 4.32 days) | 345 (75-1119) |
| 3[a] | 6/21 (29%) | 19 (73%) | 1.06 (0.46-12.15) | 8 (31%) | 18.5 (6–208 or 3.5 hours) | 437.5 (180-881) |
| 4 | 5/19 (26%) | 18 (69%) | 2.13 (0.68-7.26) | 8 (31%) | 19 (5–1514 or 1.05 days) | 421 (87-983) |

[a]The third EW session for one participant was not recorded due to computer error.

[b]Participants who received at least one reminder (percentages were derived by dividing by the number of participants who were sent the writing task).

($M$ = 728.90, $SD$ = 1057.59). Results of the effects of time, condition, and their interaction for each outcome are provided in S5 File. Findings indicate that accounting for the number of completed sessions or the length of EW content does not qualitatively change our results; there was no evidence of a condition by time interaction for any outcomes, suggesting that changes in outcomes across time were not different between the EW and control groups.

**Experience**

A subset of 11 EW participants (41%) completed qualitative interviews. Their demographic and clinical characteristics are provided in Table 1. They were mostly women, self-identifying as white, and residing in high-income neighbourhoods. Most of this subset of participants had a diagnosis of MDD ($n$ = 9, 81%), and the rest of the participants had PDD ($n$ = 2, 18%), with other comorbid conditions including GAD ($n$ = 4, 36%) and social anxiety ($n$ = 3, 27%). Five participants (45%) were taking psychotropic medications during the study, five (45%) completed psychotherapy prior to participating, and one (9%) was in psychotherapy during the study. All 11 participants completed four sessions of EW and writing times across all sessions ranged from 5 to 208 minutes (Median = 21 minutes). Their EW text lengths ranged from 87 to 1,219 words (Median = 448.5).

The participants generally reported the intervention to be feasible. Few participants ($n$ = 2, 18%) reported barriers to completing the writing task on their computers, tablets, or phones (i.e., due to not having access to their personal computer or experiencing a technical issue with the intervention platform). Most participants ($n$ = 9, 82%) also expressed that the EW instructions were clear and easy to follow. Many ($n$ = 10, 91%) reported it was easy to be open and candid in expressing their thoughts and feelings, with one noting it was helpful to write from home "in a private and safe place". However, some participants reported issues completing EW as instructed. Common reasons were forgetfulness, procrastination, and difficulties finding the time or place to write. For example, one participant reported often not completing the task due to their symptoms, which presented "issues with focus, concentration and memory". Other participants ($n$ = 4, 36%) described feeling unmotivated, or tendencies to procrastinate (e.g., by "putting things to do in front of the writing task"). Another participant described an "emotional barrier", or an unwillingness "to get into an emotional 20 minutes". Nevertheless, many participants ($n$ = 7, 64%) reported that the writing task was engaging, and the 5 participants (45%) who received text reminders reported finding them useful.

Themes related to the perceived utility of EW emerging from our analysis, and select paraphrased statements from participants, are provided in Table 3. Participants could be classified in two groups based on the themes "Emotional offloading or release" and "Cognitive processing or perspective". Seven participants (64%) found some value in EW related to one or both of these themes (i.e., the value group); four participants (36%) did not report any such benefits (i.e., the no-value group), generally finding EW to be unproductive or even harmful (see theme "Cognitive pitfalls"). Mean PHQ-9 scores at each time point for these participants, stratified by group membership, are depicted in S6 File. All participants in the no-value group, as well as some participants in the value group, ($n$ = 8, 73%) reported negative feelings during EW, which sometimes lingered after writing (see "Emotional re-experiencing"). Feelings were often attributed to recalling or re-experiencing past thoughts and emotions, and included feeling annoyed, frustrated, agitated, sad, and uncomfortable. Despite negative emotions, all participants in the value group reported some degree of "Emotional offloading or release", in that EW provided an outlet for "throwing off" or offloading negative emotions, so that these emotions would not come up at less opportune times (Table 3). All but one participant in the value group additionally reported effects related to "Cognitive processing and perspective", which involved EW offering "clarity", a gain in perspective, or enabling participants to reconceptualize their challenges. Other benefits related to this theme involved EW helping participants to "subjectively organize [their] reactions" or "dig deeper" into their feelings, reflecting on their causes or appropriateness (Table 3). Although participants across both groups acknowledged that EW would not solve their problems, participants in the no-value group also reported EW to be unproductive (see "Cognitive Pitfalls"). For example, EW took them to "a negative place", got them "stuck in a negative frame of mind", or served as a reminder that they could not fix their problems

(Table 3). Accordingly, all but one participant in the no-value group expressed that they were not motivated to complete EW, reporting that they would procrastinate, not complete it daily, or find later sessions to get tedious. One participant mentioned a general malaise related to their depression, making it "hard to get going", which carried over into EW. Some of these experiences were common across participants in both groups. For instance, participants in the value group reported experiences consistent with "Cognitive Pitfalls", despite other benefits related to emotional offloading or cognitive processing.

Participants provided suggestions to improve EW, which were primarily related to guidance and timing. Although some participants reported enjoying the freedom to write about whatever they wanted, six (55%) expressed a desire for more structured or detailed instructions. For example, instructions could guide writers through a progressive exploration of different topics or experiences, as well as potential ways to approach personal problems. Another participant suggested reviewing and reflecting on past EW entries, perhaps with feedback from a therapist or counsellor. Five participants

**Table 3. Themes, definitions, and select quotes from the analysis of qualitative interviews.**

| Theme[a] | Definition | Select paraphrased statements |
|---|---|---|
| Emotional re-experiencing | References to recalling or re-experiencing unpleasant emotions, thoughts, or experiences, which contribute to unpleasant emotions felt during and/or after EW (reported by participants in the value group and no-value group). | "I didn't like having to conjure up things in my life which are painful, it was hard. It didn't go the way I thought it would - I thought getting it out would make me feel better. But I mostly felt like I was just bringing up things there were hard." <br> "When I was writing and thinking and focusing on what has been causing my depression, I had those feelings of being uncomfortable and sad during writing." <br> "When you remember something, you're feeling it again … those negative feelings stuck around after." <br> "I would get frustrated about the topic, agitated, thinking about the issue." |
| Emotional offloading or release | References to EW being cathartic or offering relief, and/or offering an outlet to offload or release unpleasant emotions or thoughts (reported by participants in the value group only). | "I knew that after I had thought about these things that bothered me, I could let it go and not think about it for the rest of the day. If I don't take time to reflect, it might hit me when I don't want to be thinking about it. It's good to get it out of the way." <br> "It was a good experience to throw off the bad emotions, sort of like talking to a friend." <br> "I'm glad I had a spot to put it down instead of leaving it in my head … it helps." <br> "As I was writing, it felt good to get things off my chest. It felt like it was done. I could remove those thoughts or at least I knew I had done something about them." |
| Cognitive processing or perspective | References to EW promoting emotional and/or cognitive processing (i.e., organizing or articulating thoughts and feelings), deeper reflection or insight, and/or cognitive reframing or gaining perspective (reported by participants in the value group only). | "It helped me to organize my thoughts. You know, when you have horrible reactions to things, that can be overwhelming, but getting it out and subjectively organizing these reactions is useful." <br> "It let me reflect more than normally where I would let it just swirl in my head. Once I started understanding more about my thoughts and what was going on, rather than just saying 'I'm sad', it was more like 'why am I sad?' … it allows you to put words to emotions that you haven't before, lets you decide on whether you should be feeling things. For example, 'should I be angry in this situation?'" <br> "I found it very helpful for me to just dig deeper than I have in the past with my emotions and feelings. In therapy, I haven't said all these things to someone, so I found it useful to express them in writing." <br> "It helped me conceptualize some of the challenges I was facing a little, I was able to frame the challenges in a way that was not self-punishing." |
| Cognitive pitfalls | References to getting stuck in negative thoughts or emotions; references to overthinking or finding EW cognitively unproductive (reported by participants in the value group and no-value group) | "Sometimes talking to someone or writing something down can be helpful, … but the flipside is that you can talk or write too much about something. The more I talk about something or write about something, the more it keeps me stuck in a negative frame of mind." <br> "It reminded me I could express what's on my mind, but I couldn't fix it" <br> "I did not find it useful … I did not learn anything new, and by the time I was getting to the point emotionally where maybe I could have some catharsis, by then, it was over." <br> "Writing by yourself is only self-thinking, for example, every day I was writing the same thing and my own perspective was the same" |

[a]Participants were classified into the 'value' group if they described an effect of EW related to themes "Emotional offloading or release" or "Cognitive processing or perspective".

(45%) expressed a desire for oversight or support, suggesting that opportunities to discuss EW with another person could encourage deeper reflection or offer perspective, thereby maximizing cognitive benefits. Eight participants (73%) reported that the time limit was constraining, making them feel pressured or worried they would not have time to delve into a given topic. One participant felt strongly that 20 minutes was not enough to achieve meaningful benefits, finding that it was "hard to get going" and only getting "into a rhythm" by the end of each session. Participants suggested offering a range for EW timing or leaving its duration open-ended.

## Discussion

Given the potential to administer EW online as a simple and accessible intervention for Depressive Disorder, we explored its feasibility, efficacy, and perceptions of its emotional and cognitive effects in a sample of 53 patients diagnosed with a Depressive Disorder (MDD or PDD), who were randomized to EW or to a control condition. EW was feasible to administer online; most participants assigned to the intervention completed at least three EW sessions, but few wrote for at least 20 minutes, and many did not write on consecutive days. Our predictions with respect to efficacy were not supported, which may be related to this variation in participants' adherence to the EW protocol. As compared to control participants, EW participants did not report reduced severity of their MDD/PDD and GAD symptoms, functional impairment, or perceived problem complexity following the intervention or at a one-month follow up. In qualitative interviews, most participants reported online EW to be feasible, and many found it helpful for offloading or releasing negative emotions or thoughts, and processing or exploring them more deeply. However, a substantial minority of participants did not report these benefits, finding EW both unpleasant and unproductive. Suggestions to improve the intervention included providing more specific writing instructions and integrating clinical oversight or opportunities to discuss EW content with a counsellor or therapist.

Our findings on feasibility are consistent other studies, which report that only about one quarter of participants assigned to online EW complete it as instructed [28]. Most EW participants completed at least three sessions, but some were as short as 5 minutes. Although EW may have been more effective if participants wrote for longer periods of time, research suggests that benefits can be achieved with sessions as short as 3–5 minutes [48]. In qualitative interviews, participants cited benefits of completing EW online (i.e., "in a private place"), suggesting that online administration was acceptable. However, many participants forgot to complete the tasks, or cited difficulties finding time or overcoming emotional barriers to its completion. Participants also reported that they would have preferred an open-ended writing duration or suggested range, since there was more to write about on some days than others. We found that writing duration tended to decrease across EW sessions over time, but interestingly, the same trend was not observed for word count (Table 2). This suggests that participants became more comfortable or expressive in later sessions, writing more content in less time; however, this experience was not reported during qualitative interviews, although some participants did report becoming more engaged in EW toward the end of each session.

Contrary to our expectations, EW did not impact symptoms of MDD/PDD or GAD following the intervention or at a one-month follow up. This finding is consistent with some studies of EW in undergraduate or non-clinical samples [18]. However, the finding is inconsistent with one prior study in which in-person EW had a large effect in reducing depressive symptoms in participants diagnosed with MDD, relative to a control condition [23]. This discrepancy in findings may be related to the method of administration. Participants completing EW in-person may have been more likely to adhere to instructions or they may have benefitted from non-specific effects of in-person participation. Studies suggest that effects of EW on symptoms of depression, anxiety, and stress are smaller when EW is completed online, as compared to in-person settings [11]. EW completed in-person is also more effective for alleviating symptoms of post-traumatic stress disorder, potentially because writing by hand elicits more emotional processing than typing [49], or individuals may be more likely to adhere to the intervention and discuss traumatic events in a clinical setting [50]. If effects of administering EW online to samples with Depressive Disorders are small, our study is underpowered to detect them (see S4 File). An interesting direction for future work therefore involves comparing outcomes between EW administered online and in-person in

samples with Depressive Disorders. Furthermore, participants varied with respect to their adherence to the EW protocol; few wrote for at least 20 minutes, and many had an average interval of one week or more between writing sessions. Given that short intervals between writing sessions (of 1–3 days) have been associated with greater benefits of EW for mental health [11], non-adherence to the EW protocol may be an alternative explanation for our findings. It is also possible that other studies involving clinical samples finding no effect of EW were not published due to publication bias or the 'file drawer effect', or the tendency for studies reporting null effects to remain unpublished [51]. At the same time, meta-analyses evaluating impact of EW on symptoms of depression or anxiety have found no evidence of publication biases in their included studies [11,18].

Alternatively, EW may not have been effective in our sample of participants, regardless of its format. According to qualitative interviews, more than a third of participants did not report the task to be engaging, which may have limited its efficacy. Some participants also found EW unproductive, not because they were not engaged in the task, but because it got them stuck in a negative place or frame of mind. This finding supports the suggestion that EW increases rumination or exacerbates negative thoughts and emotions, perhaps because it offers little guidance for processing difficult material [26,27]. Some of our participants expressed a desire for more guidance or an opportunity to convene with a counsellor or therapist. Even some participants who found value in EW offered this suggestion, acknowledging that writing on one's own can be redundant, or a form of "self-thinking". Overall, findings from qualitative interviews may suggest that EW decreased symptoms in some participants (i.e., in the value group) and increased them in others (i.e., in the no value group). Although we did not pursue this analysis formally due to our small sample size, mean PHQ-9 scores for these two groups are depicted in S6 File. On average, symptoms decreased over time in both groups, but participants in the no-value group experienced temporary increases in symptoms following EW. Given efforts to identify moderators of EW impacts [52], reported value as a moderator of its effect on depressive symptoms may be a future direction for this work. Another potential moderator is the presence of cognitive symptoms (e.g., "issues with focus, concentration, or memory" as noted by one participant), which may present challenges to cognitive processing or deriving value from the intervention.

In qualitative interviews, participants reported emotional and cognitive effects related to EW that are consistent with its hypothesized therapeutic mechanisms. Most participants reported negative feelings during and after writing. Emotional and cognitive processing related to benefits of EW [10,14] may rely on the emergence of negative emotions while writing about a difficult personal problem or stressor [42]. Although participants reported that writing about negative topics contributed to negative emotions (Table 3), studies find that this effect is temporary [12,53]. These emotions may contribute to a better understanding of events, changes in perspective, or other relevant insights, which are thought to underlie its benefits for psychological wellbeing [10]. For example, in another study of online EW, the use of words related to negative emotion or cognitive processing were the most important predictors of subsequent decreases in depressive symptoms [54]. Heightened symptoms during periods of cognitive processing related to other psychotherapeutic interventions have also been associated with better outcomes for MDD in the long-term [13,55]. Although most participants in our study reported similar benefits related to emotional and cognitive processing (Table 3), these benefits did not translate to changes in symptom scores, functional impairment, or even perceived problem complexity. This paradoxical effect has been observed in prior work. One recent study did not find changes in symptoms of depression or anxiety related to online EW about stressors related to COVID-19, even though most participants who took part in qualitative interviews reported that the experience of writing was useful and personally meaningful [29]. A similar effect was noted in another study of EW in adolescents, leading to the suggestion that, even if EW does not result in measurable symptomatic improvements, the insights and benefits reported by participants should not be overlooked [56].

In qualitative interviews, participants also offered several suggestions on improving EW, which may inform the development of online protocols in future research, potentially involving co-design with individuals with lived experience of Depressive Disorder. Many participants expressed a desire for more specific instructions or guidance, which is consistent with prior work [18] showing that more specific EW instructions are associated with greater reductions in depressive symptom

severity, potentially because this makes it easier to adhere to EW requirements [27,52]. To evaluate this possibility, future studies may involve tailoring EW instructions to promote emotional or cognitive processing of a specific concern or negative life event (e.g., financial difficulties, relationship dissolution). Additionally, some participants expressed that the number or duration of writing sessions was not sufficient to yield meaningful benefits. This suggestion is consistent with findings that more sessions of EW are associated with larger improvements in depressive symptoms [15,18]. Thus, modifying EW to include more specific instructions over a longer period of time (e.g., 30 or more days) [15] may increase its efficacy. However, it is unclear whether such an intervention would benefit all participants (e.g., those in the no-value group), highlighting a need for research into this modified and extended version of online EW, potentially with oversight or clinical support for participants that find writing unhelpful. Future research may also involve integrating opportunities into the EW protocol to convene with a counsellor or therapist. This modification may increase engagement and adherence when EW is administered online, and it may facilitate cognitive and emotional processing, particularly for participants with challenging cognitive symptoms or if participants find themselves getting "stuck" in their negative thoughts. Given recent advancements in generative artificial intelligence (AI) [57], capabilities of AI-based conversational agents for this purpose might also be explored.

## Limitations

Our findings should be interpreted in the context of several limitations. First, because participants who could not be reached after two reminders or contact attempts were assumed to have dropped out of the study, we could not capture their outcomes or experiences. Participants who dropped out of the study before completing EW may have been less likely to benefit from it, potentially biasing results. Findings from qualitative interviews may also be biased for the same reason: participants who dropped out of the study before completing EW may not have found it to be helpful, making them less willing to share their experiences. Our qualitative analysis may therefore offer a limited understanding of the negative impacts of EW. Our randomization was also not entirely successful; participants assigned to complete EW had more individuals in the moderately-high income category, as compared to those in the control group. However, this measure reflects area-level or neighbourhood income, which may not map on to individual, household incomes in mixed-income communities. Given our focus on feasibility, our inclusion criteria were broad and included participants with co-morbid conditions and those taking psychotropic medications or undergoing psychotherapy. Due to our small sample size, we did not control for these clinical characteristics in our analysis, and they may have impacted outcomes. Over half of participants in our sample were taking psychotropic medications and many had either just completed or were undergoing group psychotherapy. We do not know whether or how these treatments (in particular, antidepressants) interfere with emotional and cognitive processing induced by EW. We did not formally assess participants for a Depressive Disorder, but we were able to confirm this diagnosis as recorded by psychiatrists in the health records of all but one participant. We also did not formally assess for severe cognitive impairment, so we do not know the extent to which symptoms like difficulties with memory or concentration impacted our feasibility findings. Finally, we did not employ a neutral, control writing task due to our focus on feasibility, so it is unclear whether findings are related to the online writing intervention or the emotional nature of the writing topic.

## Conclusions

Overall, our findings suggest that although EW can be feasibly administered online to patients with Depressive Disorders, it was not associated with measurable reductions in symptoms of MDD/PDD or GAD, functional impairment, and perceived complexity of personal problems. Thus, administering EW online as deployed in the current study may not be helpful in patients diagnosed with Depressive Disorders. Given benefits related to emotional and cognitive processing reported by some participants, it may be possible to improve the EW protocol for evaluation in future research, such as to offer more sessions with specific instructions or guidance [15,18] and with opportunities to discuss writing content with a coach or therapist.

## Supporting information

**S1 File. Study protocols and checklists.**
(DOCX)

**S2 File. Expressive writing interview guide.**
(DOCX)

**S3 File. Clinical characteristics extracted from health records.**
(DOCX)

**S4 File. Post hoc power analyses.**
(DOCX)

**S5 File. Per protocol analyses: Results.**
(DOCX)

**S6 File. Depression severity stratified by value group.**
(DOCX)

## Author contributions

**Conceptualization:** Marta M. Maslej, Abigail Ortiz, Paul W. Andrews, Benoit H. Mulsant.

**Data curation:** Marta M. Maslej, Abigail Ortiz.

**Formal analysis:** Marta M. Maslej.

**Funding acquisition:** Marta M. Maslej.

**Methodology:** Marta M. Maslej, Abigail Ortiz, Paul W. Andrews, Benoit H. Mulsant.

**Project administration:** Marta M. Maslej.

**Resources:** Benoit H. Mulsant.

**Supervision:** Benoit H. Mulsant.

**Writing – original draft:** Marta M. Maslej.

**Writing – review & editing:** Abigail Ortiz, Paul W. Andrews, Benoit H. Mulsant.

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
