## [Decision Letter · Decision Letter 0]

PMEN-D-24-00481

Feasibility, efficacy, and perceptions of an online writing intervention in patients with Major Depressive Disorder: A randomized, multi-methods study

PLOS Mental Health

Dear Dr. Mulsant,

Thank you for submitting your manuscript to PLOS Mental Health. After careful consideration, we feel that it has merit but does not fully meet PLOS Mental Health’s publication criteria as it currently stands. Therefore, we invite you to submit a revised version of the manuscript that addresses the points raised during the review process.

This study has merit and, after careful evaluation, I have considered that the manuscript requires Major Revision before further consideration for publication. Please address the reviewers' comments and provide a detailed response letter outlining the revisions made.

We look forward to receiving your revised manuscript.

Kind regards,

Ariel Soares Teles

Academic Editor

PLOS Mental Health

Journal Requirements:

https://journals.plos.org/mentalhealth/s/figures 

https://journals.plos.org/mentalhealth/s/figures#loc-file-requirements 

Additional Editor Comments (if provided):

Authors are recommended to consider the issues mentioned in the reviewers' comments carefully.

Reviewers' comments:

Reviewer's Responses to Questions

**Comments to the Author**

1. Does this manuscript meet PLOS Mental Health’s publication criteria? Is the manuscript technically sound, and do the data support the conclusions? The manuscript must describe methodologically and ethically rigorous research with conclusions that are appropriately drawn based on the data presented.

Reviewer #1: Yes

Reviewer #2: Yes

2. Has the statistical analysis been performed appropriately and rigorously?

Reviewer #1: Yes

Reviewer #2: I don't know

3. Have the authors made all data underlying the findings in their manuscript fully available (please refer to the Data Availability Statement at the start of the manuscript PDF file)?

Reviewer #1: Yes

Reviewer #2: No

4. Is the manuscript presented in an intelligible fashion and written in standard English?

Reviewer #1: Yes

Reviewer #2: No

5. Review Comments to the Author

Reviewer #1: Important note: This review pertains only to ‘statistical aspects’ of the study and so ‘clinical aspects’ [like medical importance, relevance of the study, ‘clinical significance and implication(s)’ of the whole study, etc.] are to be evaluated [should be assessed] separately/independently. Further please note that any ‘statistical review’ is generally done under the assumption that study specific methodological [as well as execution] issues are perfectly taken care of by the investigator(s). This review is not an exception to that and so does not cover clinical aspects {however, seldom comments are made only if those issues are intimately / scientifically related & intermingle with ‘statistical aspects’ of the study}. Agreed that ‘statistical methods’ are used as just tools here, however, they are vital part of methodology [and so should be given due importance]. I look at the manuscript in/with statistical view point, other reviewer(s) look(s) at it with different angle so that in totality the review is very comprehensive. However, there should be efforts from authors side to improve (may be by taking clues from reviewer’s comments). Therefore, please do not limit the revision only (with respect) to comments made here.

COMMENTS: I have different opinion/views/observations/concerns or rather questions regarding a few issues which are given below:

Firstly, although the title [Feasibility, efficacy, and perceptions of an online writing intervention in patients with Major Depressive Disorder: A randomized, multi-methods study] indicates/clarifies that it is a ‘Feasibility’ study only, it is better to add word ‘pilot’ in the title as when the study is ‘pilot’ in nature, many things are ignored [loosely looked at / evaluated; and that more latitude (i.e., leeway, freedom, liberty) in statistical requirements is observed/given] {example: sample size is not a big issue, other methodology issues need not be looked at very rigorously, following of CONSORT guidelines (in case of RCTs) is not strictly observed}.

I noted that your ABSTRACT is well drafted (in my opinion), but is ‘assay type’. It is preferable [refer to item 1b of CONSORT checklist 2010: Structured summary of trial/study design, methods, results, and conclusions] to divide the ABSTRACT with small sections like ‘Objective(s)’, ‘Methods’, ‘Results’, ‘Conclusions’, etc. which is an accepted practice of most of the good/standard journals [including this one, though the ‘Guidelines to Authors’ might/did not specify an Abstract format, it is desirable]. It will definitely be more informative then, I guess, whatever the article type may be {though Section headings may differ for different Article Types [example: Study Protocol]}.

As stated in lines 398-400 that ‘According to a post-hoc power analysis (reported in Supplement S4), our sample size of 23 participants per condition was powered to detect large effects (f=0.47)’ sample size (of 23 participants per condition) will suffice to detect large effects only. Whereas in such studies the ‘effect size’ observed/found generally is very ‘small’ [requiring a large sample size]. According to table-2 on page 158 of Jacob Cohen’s paper “A power primer” in Psychological Bulletin, 1992, vol.:112, pp 155-159 [which is a sort of summary of the excellent book by Cohen himself titled ‘Statistical power analysis for the behavioral sciences’, Academic Press, 1977, New York] even for medium effect size you need n=64 per group (type-I error=0.05, power=80%). Please note that the ‘effect size’ assumed should have some basis (exact reference needs to be quoted) or reasonable/realistic, else the study is very likely ‘not to be able to’ detect a difference despite its presence.

Very good that this fact (small sample size & consequences) is mentioned at few places in the manuscript [examples: line 553 ‘Although we did not pursue this analysis formally due to our small sample size, mean PHQ-9….’ and lines 618-620 ‘Due to our small sample size, we did not control for these clinical characteristics in our analysis, and they may have impacted outcomes’]. When the outcome is measured in ‘ordinal’ level of measurement then one has to inflate sample size by 10 to 20% and also then the application of suitable non-parametric (or distribution free) test(s) is/are indicated/advisable [even if distribution may be ‘Gaussian’ (also called ‘normal’)]. Though the measures/tools used are appropriate, most of them [examples: Patient Health Questionnaire (PHQ-9), Generalized Anxiety Disorder (GAD-7), WHO-Disability Assessment Schedule (WHODAS), Problem Complexity Questionnaire (PCQ)] are likely to yield data that are in [at the most] ‘ordinal’ level of measurement [and not in ratio level of measurement for sure {as the score two times higher does not indicate presence of that parameter/phenomenon as double (for example, a Visual Analogue Scales VAS score or say ‘depression’ score)}].

The fact brought-out in lines 141-2 {benefits of EW in individuals with MDD may not only reflect changes in symptoms, but they may extend to other outcomes} and in lines 155-7 {it is possible that in studies not reporting statistically significant effects of EW on MDD [19], participants might nevertheless experience benefits related to other aspects of their lives} is appreciated as this is the reality.

As pointed out in ‘important note’ above “This review pertains only to ‘statistical aspects’ of the study and so ‘clinical aspects’ should be assessed separately/independently [one should carefully consider/look at the clinical implications of the study]. In my opinion, to make this article acceptable (which is quite possible and easy), a small amount of re-vision (re-drafting) may be needed. Therefore, only ‘Minor revision’ is recommended. However, please do not limit the revision only (with respect) to comments made here.

Reviewer #2: The introduction often drifts into providing general background information on MDD and the COVID-19 pandemic’s effects on mental health. This inclusion feels somewhat disconnected from the core objective of the study. A clearer articulation of the specific gap the current study aims to fill would make the introduction more focused and compelling.

The reliance on meta-analyses to report mixed findings of EW’s effects on MDD is important but feels somewhat excessive. The argument for why these mixed results matter in the context of your study is underdeveloped. Rather than only presenting findings, the paper should more directly critique how existing studies' limitations (e.g., homogeneous samples of college students, methodological differences) might affect the applicability of the results to clinical populations.

The rationale behind the decision not to include a control writing task is not fully convincing. The justification appears to be based on a desire to focus on feasibility, but the lack of a well-defined control condition is a significant methodological weakness.

While the hypotheses presented toward the end of the section are clear, they are somewhat redundant with earlier statements. Repetition of expected outcomes could be avoided by refining the hypotheses and linking them more clearly to the study’s specific aims.

There are moments where the language is overly dense or technical, making it challenging to follow the logical flow.

While participants were excluded if they had cognitive impairments that might hinder completing the writing task, there is no clear justification for why cognitive impairments, particularly those relevant to MDD (such as memory or concentration difficulties), would not be formally assessed during enrollment. This criterion should have been assessed systematically rather than left to clinician discretion, as cognitive impairments are common in patients with MDD. This omission creates a potential confound in the study sample that could influence the feasibility outcomes.

Although the study mentions that clinicians were asked not to refer patients with suicidal thoughts or behaviors, there is no formal assessment of this criterion at enrollment. This is a significant oversight, particularly in a population with MDD, where suicidal ideation is prevalent. The lack of a standardized screening process for this risk factor leaves the study vulnerable to potential ethical and safety concerns, especially when administering the intervention remotely. It is essential that a clearer protocol for identifying and excluding at-risk participants is included.

Excluding participants based on the ability to understand, speak, or write in English should be better justified. In a multicultural context like Canada, where diverse linguistic populations are present, this may inadvertently exclude a significant portion of the population who could benefit from the intervention.

The justification for using postal codes should be more detailed, and it would be helpful to discuss potential limitations of this approach.

The PCQ rationale and validity need further clarification. Furthermore, the use of a single item per domain (e.g., “These problems have left me in a dilemma”) may oversimplify complex personal issues, and a justification for the use of this approach, as opposed to a more comprehensive measure, is needed.

The control group, which did not complete any writing task, is critical to assess the efficacy of the writing intervention. However, the justification for this control condition is weak. The section mentions that the aim was to focus on feasibility, but this raises the question of whether omitting a proper control group could confound the study's ability to draw valid conclusions about the efficacy of the intervention.

The instructions for the writing task are somewhat vague in places, particularly in terms of guiding participants on how to reflect on and process their emotions. The phrase "write your very deepest thoughts and feelings" is potentially overwhelming for individuals with MDD, as it may prompt emotional flooding rather than therapeutic processing. There is no discussion of how the task was adapted for the target population or how it might be modified for individuals who experience higher distress or difficulty engaging. Further elaboration on how the writing prompt was designed to be both therapeutic and manageable is needed.

The follow-up procedures appear to be standard, but the criteria for determining dropout versus non-completion are underexplained. Participants who did not respond after two reminders were assumed to have dropped out, but what was done with those who responded late or partially completed the intervention is unclear. There is no discussion of how partial participation was handled in the analysis, which could impact the interpretation of feasibility and efficacy.

There is no clear explanation of how missing data will be handled, especially in the case of dropouts or partial responses. Given the challenges with participant engagement in online interventions.

What is the rationale for selecting these models over simpler alternatives (e.g., repeated measures ANOVA)? A brief discussion on why mixed effect models are the most suitable choice for the data structure.

In the per-protocol analysis, there is a focus on the number of completed sessions and text length as predictors, but there is no mention of how these variables will be interpreted in the context of clinical significance. While statistical significance is important, it would be useful to link these measures to real-world improvements in patient outcomes.

The qualitative data section could benefit from more detail on how the analysis will be conducted.

The sample size (63 participants) is relatively small, and the significant attrition (from 63 to 46 participants) needs more in-depth discussion. Why did nearly 27% of participants drop out?

Are the results generalizable given the significant loss to follow-up?

The use of abbreviations like "EW" and the inconsistent application of full terms and abbreviations may confuse readers.

There is some redundancy in the reporting of demographic characteristics.

The authors fail to discuss the implications of the statistical results in sufficient depth. Are there alternative explanations, such as the low engagement with the EW task or methodological flaws, that could account for these results?

Did the low engagement levels contribute to the null findings?

Clarify how the power analysis was conducted and why the sample size of 23 participants per condition was considered adequate?

Why was there such variation in completion rates and writing duration? How can the authors account for the skewed distribution of session completion, with many participants failing to write for at least 20 minutes per session?

What changes could be made to make EW more effective, and how might these suggestions influence the design of future trials?

The authors provide useful suggestions for improving the EW protocol, including offering more specific instructions and integrating clinical oversight. However, these suggestions seem to assume that the EW protocol can be improved by simply making it more structured or personalized, but it’s not clear if this would address the underlying limitations of the intervention.

The writing text feels overly dense and complex, particularly when discussing findings from previous studies and the nuances of participant feedback.

6. PLOS authors have the option to publish the peer review history of their article (what does this mean?). If published, this will include your full peer review and any attached files.

**Do you want your identity to be public for this peer review?** For information about this choice, including consent withdrawal, please see our Privacy Policy.

Reviewer #1: No

Reviewer #2: No

---

## [Decision Letter · Decision Letter 1]

PMEN-D-24-00481R1

Feasibility, efficacy, and perceptions of an online writing intervention in patients with Major Depressive Disorder: A randomized, multi-methods pilot study

PLOS Mental Health

Dear Dr. Mulsant,

Thank you for submitting your manuscript to PLOS Mental Health. After careful consideration, we feel that it has merit but does not fully meet PLOS Mental Health’s publication criteria as it currently stands. Therefore, we invite you to submit a revised version of the manuscript that addresses the points raised during the review process.

Authors should consider the specific comments from the reviewer, which consider important aspects of the manuscript/study.

We look forward to receiving your revised manuscript.

Kind regards,

Ariel Soares Teles

Academic Editor

PLOS Mental Health

Journal Requirements:

Additional Editor Comments (if provided):

Reviewers' comments:

Reviewer's Responses to Questions

**Comments to the Author**

1. If the authors have adequately addressed your comments raised in a previous round of review and you feel that this manuscript is now acceptable for publication, you may indicate that here to bypass the “Comments to the Author” section, enter your conflict of interest statement in the “Confidential to Editor” section, and submit your "Accept" recommendation.

Reviewer #1: All comments have been addressed

Reviewer #3: All comments have been addressed

2. Does this manuscript meet PLOS Mental Health’s publication criteria? Is the manuscript technically sound, and do the data support the conclusions? The manuscript must describe methodologically and ethically rigorous research with conclusions that are appropriately drawn based on the data presented.

Reviewer #1: (No Response)

Reviewer #3: Yes

3. Has the statistical analysis been performed appropriately and rigorously?

Reviewer #1: (No Response)

Reviewer #3: Yes

4. Have the authors made all data underlying the findings in their manuscript fully available (please refer to the Data Availability Statement at the start of the manuscript PDF file)?

Reviewer #1: (No Response)

Reviewer #3: Yes

5. Is the manuscript presented in an intelligible fashion and written in standard English?

Reviewer #1: (No Response)

Reviewer #3: Yes

6. Review Comments to the Author

Reviewer #1: (No Response)

Reviewer #3: Thank you for inviting me to review this manuscript. Overall, I found this manuscript well written and interesting. I particularly liked the inclusion of the qualitative data and the opportunities it provided for participants to make recommendations for intervention improvements.

A major question I have about this study is in relation to the focus and inclusion criteria. It states in the title that the focus is patients with MDD, and this is included as an inclusion criterion (i.e., formal diagnosis of MDD from a psychiatrist), but unless I am mistaken only 71% of participants have a diagnosis of MDD with a further 24% having a diagnosis Persistent depressive disorder/Dysthymia. Therefore, the inclusion criteria, title and other parts of this manuscript do not reflect the clinical characteristics of the participants.

Please can you clarify how participants without a MDD diagnosis were recruited if this was part of the inclusion criteria (i.e., a diagnosis of MDD from a psychiatrist)?

Given that only 71% of the participants had a diagnosis of MDD, I would consider removing any mentions that the findings presented in this manuscript are specific to MDD only to perhaps "mental health outpatients"? It appears that only 48 of the participants who could have their diagnoses retrived from the EHR (i.e., 51) had a recorded diagnosis of a depressive disorder (e.g., either MDD or Persistent depressive disorder/Dysthymia) and instead their primary disorder may be non-mood related.

In this case it may also be worthwhile to add to your manscript literature examining the benefits of EW for disorders other than depressive disorders as your study appears to also examine due to the clinical characteristics of the sample.

Abstract

In the abstract, I would consider providing the number of participants who participated in the interview.

You state that 54 participants were included in the study, but it would be helpful to clarify that this is baseline data and that 46 participants completed the study.

Introduction

The introduction very clearly sets out the need for identification of scalable interventions.

In lines 83 and 88 where you have mentioned meta-analyses by specific authors, you may want to provide the year of publications of these meta-analyses.

On line 108 you mention that online delivery of EW is potentially unhelpful, but it is not clear in the preceding section whether this delivery modality was explored in any of the studies cited. I would suggest removing the word “online” from this sentence.

Very strong justification of supplementing quantitative data with qualitative data.

Materials and Methods

If the diagnosis of MDD was retrieved from the EHR was it possible to confirm that this was a current diagnosis and not a historic one, as this may have implications for the description of the participants. Additionally, please can you confirm that prior to beginnning CBT with a psychotherapist that all of the participants recruited through this pathway had also been assessed by a psychiatrist as per the inclusion criteria? And if so, was any data available on the time elapsed between the diagnosis from the psychiatrist and enrollment in the study as their symptoms may have improved in this time.

Was there any co-design of the EW protocol with the population of interest (e.g., those with MDD or clinicians)? It would be helpful to know this to understand elements of the study such as the frequency of EW chosen.

In the ethics statement and data availability section, do you have an ethics approval number?

In the descriptive section of the statistical analysis section, please could you clarify what you mean by recoding the racial ethic categories into a binary variable? What were the two binary groups? I see this is described in table 1 but may be helpful to also state in the methods.

As part of feasibility and efficacy, did you collect any data about the reminders (e.g., the rate of response following the reminder 1 or 2)? This may be good to report for future studies which may build off your methods/findings if this is an important tool for encouraging engagement, especially as the reminders were reported to be useful in the qualitative data.

Please confirm how codes were identified in the qualitative data. Was this done by a single member of the research team, or was it multiple under blinded conditions and then compared?

Results

I wonder if it would be possible to present the engagement data (i.e., number of sessions completed) as a figure? (e.g., a Sankey)? It is a little hard to follow the engagement when it is written.

It would be helpful to also add the intervals between sessions of EW to Table 2 if possible.

I would consider updating Table 3 to better reflect the “value” and “no value” groups as you define them. It isn’t very easy to quickly determine how these themes/quotes relate to each group. This would be helpful particularly as you are using these groups to structure the way you have written this section.

Discussion

Line 498 onwards you hypothesize about why the intervention did not impact MDD/GAD symptoms. Is it possible that the length of the intervention/study was too short for it to have a significant impact, with it only being 4 opportunities to engage in EW offered across 4 days?

Is it possible that there are some cognitive symptoms associated with MDD which would make certain individuals inappropriate for EW as an intervention, as is also reflected in your qualitative data? Should this be considered in future research examining the efficacy of EW?

Limitations

Why in particular antidepressants? Surely psychotherapy is more likely to impact on EW with this potentially improving the skills potentially required to engage in EW?

Conclusions

Line 610: I’d consider replacing “in its current state” with “as deployed in the current study” given your limitations.

7. PLOS authors have the option to publish the peer review history of their article (what does this mean?). If published, this will include your full peer review and any attached files.

**Do you want your identity to be public for this peer review?** For information about this choice, including consent withdrawal, please see our Privacy Policy.

Reviewer #1: **Yes: **Dr. Sanjeev Sarmukaddam

Reviewer #3: No

---

## [Decision Letter · Decision Letter 2]

Feasibility, efficacy, and perceptions of an online writing intervention in patients with Depressive Disorders: A randomized, multi-methods pilot study

PMEN-D-24-00481R2

Dear Dr. Mulsant,

We are pleased to inform you that your manuscript 'Feasibility, efficacy, and perceptions of an online writing intervention in patients with Depressive Disorders: A randomized, multi-methods pilot study' has been provisionally accepted for publication in PLOS Mental Health.

Best regards,

Ariel Soares Teles

Academic Editor

PLOS Mental Health

Reviewer Comments (if any, and for reference):

Reviewer's Responses to Questions

**Comments to the Author**

1. If the authors have adequately addressed your comments raised in a previous round of review and you feel that this manuscript is now acceptable for publication, you may indicate that here to bypass the “Comments to the Author” section, enter your conflict of interest statement in the “Confidential to Editor” section, and submit your "Accept" recommendation.

Reviewer #3: All comments have been addressed

2. Does this manuscript meet PLOS Mental Health’s publication criteria? Is the manuscript technically sound, and do the data support the conclusions? The manuscript must describe methodologically and ethically rigorous research with conclusions that are appropriately drawn based on the data presented.

Reviewer #3: Yes

3. Has the statistical analysis been performed appropriately and rigorously?

Reviewer #3: Yes

4. Have the authors made all data underlying the findings in their manuscript fully available (please refer to the Data Availability Statement at the start of the manuscript PDF file)?

Reviewer #3: No

5. Is the manuscript presented in an intelligible fashion and written in standard English?

Reviewer #3: (No Response)

6. Review Comments to the Author

Reviewer #3: Thank you very much for responding to my comments, I appreciate the clarification provided.

7. PLOS authors have the option to publish the peer review history of their article (what does this mean?). If published, this will include your full peer review and any attached files.

**Do you want your identity to be public for this peer review?** For information about this choice, including consent withdrawal, please see our Privacy Policy.

Reviewer #3: No
